# Learning Invariances in Neural Networks

**Gregory Benton**    **Marc Finzi**    **Pavel Izmailov**    **Andrew Gordon Wilson**
Courant Institute of Mathematical Sciences
New York University

## Abstract

Invariances to translations have imbued convolutional neural networks with powerful generalization properties. However, we often do not know a priori what invariances are present in the data, or to what extent a model should be invariant to a given symmetry group. We show how to *learn* invariances and equivariances by parameterizing a distribution over augmentations and optimizing the training loss simultaneously with respect to the network parameters and augmentation parameters. With this simple procedure we can recover the correct set and extent of invariances on image classification, regression, segmentation, and molecular property prediction from a large space of augmentations, on training data alone.

## 1 Introduction

The ability to learn constraints or symmetries is a foundational property of intelligent systems. Humans are able to discover patterns and regularities in data that provide compressed representations of reality, such as translation, rotation, intensity, or scale symmetries. Indeed, we see the value of such constraints in deep learning. Fully connected networks are more flexible than convolutional networks, but convolutional networks are more broadly impactful because they enforce the *translation equivariance* symmetry: when we translate an image, the outputs of a convolutional layer translate in the same way [24, 7]. Further gains have been achieved by recent work hard-coding additional symmetries, such as rotation equivariance, into convolutional neural networks [e.g., 7, 41, 44, 31]

But we might wonder whether it is possible to *learn* that we want to use a convolutional neural network. Moreover, we typically do not know which constraints are suitable for a given problem, and to what extent those constraints should be enforced. The class label for the digit '6' is rotationally invariant up until it becomes a '9'. Like biological systems, we would like to automatically discover the appropriate symmetries. This task appears daunting, because standard learning objectives such as maximum likelihood select for flexibility, rather than constraints [29, 32].

In this paper, we provide an extremely simple and practical approach to automatically discovering invariances and equivariances, *from training data alone*. Our approach operates by learning a distribution over augmentations, then training with augmented data, leading to the name *Augerino*. Augerino (1) can learn both invariances and equivariances over a wide range of symmetry groups, including translations, rotations, scalings, and shears; (2) can discover partial symmetries, such as rotations not spanning the full range from $[-\pi, \pi]$; (3) can be combined with any standard architectures, loss functions, or optimization algorithm with little overhead; (4) performs well on regression, classification, and segmentation tasks, for both image and molecular data.

To our knowledge, Augerino is the first approach that can learn symmetries in neural networks from training data alone, without requiring a validation set or a special loss function. In Sections 3-5 we introduce Augerino and show why it works. The accompanying code can be found at https://github.com/g-benton/learning-invariances.

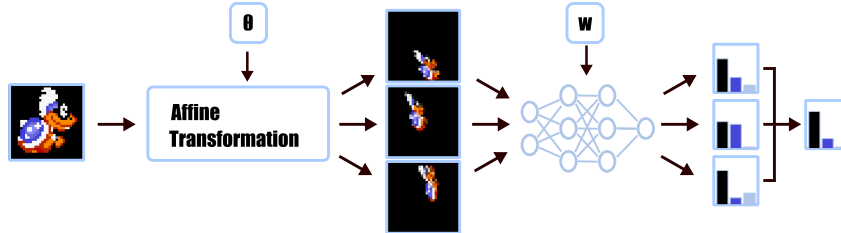

Figure 1: The Augerino framework. Augmentations are sampled from a distribution governed by parameters $\theta$, and applied to an input to produce multiple augmented inputs. These augmented inputs are then passed to a neural network with weights $w$, and the final prediction is generated by averaging over the multiple outputs. Augerino discovers invariances by learning $\theta$ from training data alone.

## 2   Related Work

There has been explosion of work constructing convolutional neural networks that have *hard-coded* invariance or equivariance to a set of transformations, such as rotation [7, 41, 44, 31] and scaling [40, 35]. While recent methods use a representation theoretic approach to find a basis of equivariant convolutional kernels [9, 41, 39], the older method of Laptev et al. [22] pools network outputs over many hard-coded transformations of the input for fixed invariances, but does not consider equivariances.

With a desire to automate the machine learning pipeline, Cubuk et al. [10] introduced *AutoAugment* in which reinforcement learning is used to find an optimal augmentation policy within a discrete search space. At the expense of a massive computational budget for the search, AutoAugment brought substantial gains in image classification performance, including state-of-the-art results on ImageNet. The AutoAugment framework was extended first to *Fast AutoAugment* in Lim et al. [27], improving both the speed and accuracy of AutoAugment by using Bayesian data augmentation [36]. Both Cubuk et al. [10] and Lim et al. [27] apply a reinforcement learning approach to searching the space of augmentations, significantly differing from our work which directly optimizes distributions over augmentations with respect to the training loss.

*Faster AutoAugment* [15], which uses a GAN framework to match augmentations to the data distribution, and *Differentiable Automatic Data Augmentation* [25] which applies a DARTS [28] bi-level optimization procedure to learn augmentation from the validation loss are most similar to Augerino in the discovery of distributions over augmentations. Both methods learn augmentations from data using the reparametrization trick; however unlike Li et al. [25] and Liu et al. [28], we learn augmentations directly from the training loss without need for GAN training or the complex DARTS procedure [28, 42, 26], and are specifically learning degrees of invariances and equivariances.

To the best of our knowledge, Augerino is the first work to *learn* invariances and equivariances in neural networks from training data alone. The ability to automatically discover symmetries enables us to uncover interpretable salient structure in data, and provide better generalization.

## 3   Augerino: Learning Invariances through Augmentation

A simple way of constructing a model invariant to a given group of transformations is to average the outputs of an arbitrary model for the inputs transformed with all the transformations in the group. For example, if we wish to make a given image classifier invariant to horizontal reflections, we can average the predictions of the network for the original and reflected input.

Augerino functions by sampling multiple augmentations from a parameterized distribution then applying these augmentations to an input to acquire multiple augmented samples of the input. The augmented input samples are each then passed through the model, with the final prediction being generated by averaging over the individual outputs. We present the Augerino framework in Figure 1.

Now, suppose we are working with a set $\mathcal{S}$ of transformations. Relevant transformations may not always form a group structure, such as rotations $R_\phi$ by limited angles in the range $\phi \in [-\theta, \theta]$. Given a neural network $f_w$, with parameters $w$, we can make a new model $\bar{f}$ which is approximately

invariant to transformations $\mathcal{S}$ by averaging the outputs over a parametrized distribution $\mu_\theta(\cdot)$ of the transformations $g \in S$:

$$\bar{f}(x) = \mathbb{E}_{g \sim \mu} f(gx). \tag{1}$$

Since the cross-entropy loss $\ell$ for classification is linear in the class probabilities, we can pull the expectation outside of the loss:

$$\ell(\bar{f}(x)) = \ell(\mathbb{E}_{g \sim \mu} f(gx)) = \mathbb{E}_{g \sim \mu} \ell(f(gx)). \tag{2}$$

As stochastic gradient descent only requires an unbiased estimator of the gradients, we can train the augmentation averaged model $\bar{f}$ *exactly* by minimizing the loss of $f(gx)$ averaged over a finite number of samples from $g \sim \mu$ at training time, using a Monte Carlo estimator.

To learn the invariances we can also backpropagate through to the parameters $\theta$ of the distribution $\mu_\theta$ by using the reparametrization trick [20]. For example, for a uniform distribution over rotations with angles $U[-\theta, \theta]$, we can parametrize the rotation angle by $\phi = \theta\epsilon$ with $\epsilon \sim U[-1, 1]$. The loss $L(\cdot)$ for the augmentation-averaged model on an input $x$ can be computed as

$$L_x(\theta, w) = \mathbb{E}_{\phi \sim U[-\theta, \theta]} \ell(f_w(R_\phi x)) = \mathbb{E}_{\epsilon \sim U[-1,1]} \ell(f_w(R_{\epsilon\theta} x)). \tag{3}$$

Specifically, during training we can use a single sample from the augmentation distribution to estimate the gradients. The learned range of rotations $[-\theta, \theta]$ would correspond to the extent rotational invariance is present in the data. With a more general set of $k$ transformations, we can similarly define a distribution $\mu_\theta(\cdot)$ over the transformation elements using the reparametrization trick $g = g_\epsilon = \epsilon \odot \theta$, with $\epsilon \sim U[-1, 1]^k$ and $\theta \in \mathbb{R}^k$. The reparameterized loss is then

$$L_x(\theta, w) = \mathbb{E}_{\epsilon \sim U[-1,1]^k} \ell(f_w(g_\epsilon x)). \tag{4}$$

In Section 3.2 we describe a parameterization of the set of affine transformations which includes translations, rotations, and scalings of the input as special cases. In this fashion, we can train both the parameters of the augmentation averaged model $\bar{f}$ consisting both of the weights $w$ of $f_w$ and the parameters $\theta$ of the augmentation distribution $\mu_\theta$.

**Test-time Augmentation**    At test time we sample multiple transformations $g \sim \mu_\theta$ and make a prediction by averaging over the predictions generated for each transformed input, approximating the expectation in Equation (1). We discuss experimental design choices for train and test time augmentation in Appendix C.

**Regularized Loss**    Invariances correspond to constraints on the model, and in general the most unconstrained model may be able to achieve the lowest training loss. However, we have a prior belief that a model should preserve *some* level of invariance, even if standard losses cannot account for this preference. To bias training towards solutions that incorporate invariances, we add a regularization penalty to the network loss function that promotes broader distributions over augmentations. Our final loss function is given by

$$L_x(\theta, w) = \mathbb{E}_{g \sim \mu_\theta} \ell(f_w(gx)) + \lambda R(\theta), \tag{5}$$

where $R$ is a regularization function encouraging coverage of a larger volume of transformations and $\lambda$ is the regularization weight (the form of $R(\theta)$ is discussed in Section 3.2). In practice we find that the choice of $\lambda$ is *largely unimportant*; the insensitivity to the choice of $\lambda$ is demonstrated throughout Sections 4 and 6 in which performance is consistent for various values of $\lambda$. This is due to the fact that there is essentially no gradient signal for $\theta$ over the range of augmentations consistent with the data, so even a small push is sufficient. We discuss further why Augerino is able to learn the correct level of invariance — *without sensitivity to $\lambda$, and from training data alone* — in Section 5.

We refer to the proposed method as *Augerino*[1]. We summarize the method in Algorithm 1.

---

**Algorithm 1:** Learning Invariances with Augerino

---

**Inputs:**
Dataset $\mathcal{D}$; parametric family $g$ of data augmentations and a distribution $\mu_\theta$ over the parameters
  $\theta$; neural network $f_w$ with parameters $w$; number $n_{\text{copies}}$ of augmented inputs to use during
  training; number of training steps $N$.
**for** $i = 1, \ldots, N$ **do**
  | Sample a mini-batch $\tilde{x}$ from $\mathcal{D}$;
  | For each datapoint in $\tilde{x}$ sample $n_{\text{copies}}$ transformations from $\mu_\theta$;
  | Average predictions of the network $f_w$ over $n_{\text{copies}}$ data transformations of $\tilde{x}$;
  | Compute the loss (5), $L_{\tilde{x}}(\theta, w)$ using the averaged predictions;
  | Take the gradient step to update the parameters $w$ and $\theta$;
**end**

---

## 3.1 Extension to Equivariant Predictions

We now generalize Augerino to problems where the targets are *equivariant* rather than invariant to a certain set of transformations. We say that target values are equivariant to a set of input transformations if the targets for a transformed input are transformed in the same way as the input. Formally, a function $f$ is equivariant to a symmetry transformation $g$, if applying $g$ to the input of the function is the same as applying $g$ to the output, such that $f(gx) = gf(x)$. For example, in image segmentation if the input image is rotated the target segmentation mask should also be rotated by the same angle, rather than being unchanged.

To make the Augerino model equivariant to transformations sampled from $\mu_\theta(\cdot)$, we can average the inversely transformed outputs of the network for transformed inputs:

$$f_{\text{aug-eq}}(x) = \mathbb{E}_{g \sim \mu} g^{-1} f(gx). \tag{6}$$

Supposing that $g$ acts linearly on the image then the model is equivariant:

$$f_{\text{aug-eq}}(hx) = \mathbb{E}_{g \sim \mu} g^{-1} f(ghx) = \mathbb{E}_{g \sim \mu} h(gh)^{-1} f(ghx) = h\mathbb{E}_{u \sim \mu} u^{-1} f(ux) \tag{7}$$
$$= h f_{\text{aug-eq}}(x) \tag{8}$$

where $u = gh$ and the distribution is right invariant: for any measurable set $S$, $\forall h \in G : \mu(S) = \mu(hS)$. If the distribution over the transformations is uniform then the model is equivariant.

## 3.2 Parameterizing Affine Transformations

We now show how to parameterize a distribution over the set of affine transformations of $2d$ data (e.g. images). With this parameterization, Augerino can learn from a broad variety of augmentations including translations, rotations, scalings and shears.

The set of affine transformations form an algebraic structure known as a Lie Group. To apply the reparametrization trick, we can parametrize elements of this Lie Group in terms of its Lie Algebra via the exponential map [13]. With a very simple approach, we can define bounds $\theta_i$ on a uniform distribution over the different exponential generators $G_i$ in the Lie Algebra:

$$g_\epsilon = \exp\left(\sum_i \epsilon_i \theta_i G_i\right) \quad \epsilon \sim U[-1, 1]^k, \tag{9}$$

where exp is the matrix exponential function: $\exp(A) = \sum_{n=0}^{\infty} \frac{1}{n!} A^n$. [2]

The generators of the affine transformations in $2d$, $G_1, \ldots, G_6$, correspond to translation in $x$, translation in $y$, rotation, scaling in $x$, scaling in $y$, and shearing; we write out these generators in Appendix A. The exponential map of each generating matrix produces an affine matrix that can be used to transform the coordinate grid points of the input like in Jaderberg et al. [18]. To ensure that

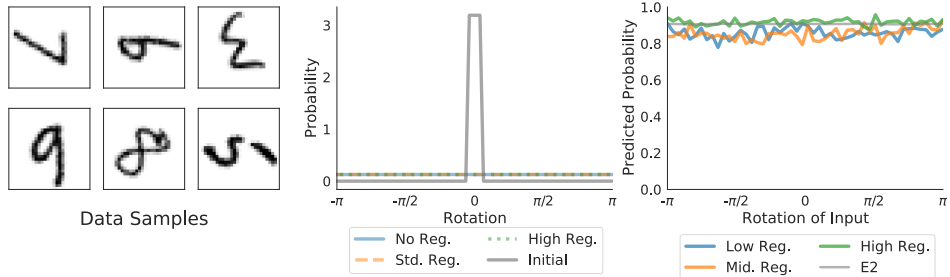

Figure 2: **Left:** Samples of the rotated digits in the data. **Center:** The initial and learned distributions over rotations. **Right:** The prediction probabilities of the correct class label over rotated versions of an image; the model learns to be approximately invariant to rotations under all levels of regularization.

the parameters $\theta_i$ are positive, we learn parameters $\tilde{\theta}_i$ where $\theta_i = \log(1 + \exp \tilde{\theta}_i)$. In maximizing the volume of transformations covered, it would be geometrically sensible to maximize the Haar measure $\mu_H(S)$ of the set of transformations $S = \exp(\text{Cube}_\theta)$ that are covered by Augerino, which is similar to the volume covered in the Lie Algebra $\text{Vol}(\text{Cube}_\theta) = \Pi_{i=1}^{k}\theta_i$. However, we find that even the negative $L_2$ regularization $R(\theta) = -\|\theta\|^2$ on the bounds $\theta_i$ is sufficient to bias the model towards invariance. More intuitively, the regularization penalty biases solutions towards values of $\theta$ which induce broad distributions over affine transformations, $\mu_\theta$.

We apply the $L_2$ regularization penalty on both classification and regression problems, using cross entropy and mean squared error loss, respectively. This regularization method is effective, interpretable, and leads to the discovery of the correct level of invariance for a wide range of $\lambda$.

## 4 Shades of Invariance

We can broadly classify invariances in three distinct ways: first there are cases in which we wish to be completely invariant to transformations in the data, such as to rotations on the rotMNIST dataset. There are also cases in which we want to be only partially invariant to transformations, i.e. *soft* invariance, such as if we are asking if a picture is right side up or upside down. Lastly, there are cases in which we wish there to be no invariance to transformations, such as when we wish to predict the rotations themselves. We show that Augerino can learn full invariance, soft invariance, and no invariance to rotations. We then explain in Section 5 why Augerino is able to discover the correct level of invariance from training data alone. Incidentally, soft invariances are the most representative of real-world problems, and also the most difficult to correctly encode a priori — where we most need to learn invariances.

For the experiments in this and all following sections we use a 13-layer CNN architecture from Laine and Aila [21]. We compare Augerino trained with three values of $\lambda$ from Equation 5; $\lambda = \{0.01, 0.05, 0.1\}$ corresponding to low, standard, and high levels of regularization. To further emphasize the need for invariance to be *learned* as opposed to just embedded in a model we also show predictions generated from an invariant $E(2)$-steerable network [9]. Specific experimental and training details are in Appendix C.

### 4.1 Full Rotational Invariance: rotMNIST

The rotated MNIST dataset (rotMNIST) consists of the MNIST dataset with the input images randomly rotated. As the dataset has an inherent augmentation present (random rotations), we desire a model that is invariant to such augmentations. With Augerino, we aim to approximate invariance to rotations by learning an augmentation distribution that is uniform over all rotations in $[0, 2\pi]$.

Figure 2 shows the learned distribution over rotations to apply to images input into the model. On top of learning the correct augmentation through automatic differentiation using *only* the training data, we achieve $98.9\%$ test accuracy. We also see the level of regularization has little effect on performance. To our knowledge, only Weiler and Cesa [39] achieve better performance on the rotMNIST dataset, using the correct equivariance already hard-coded into the network.

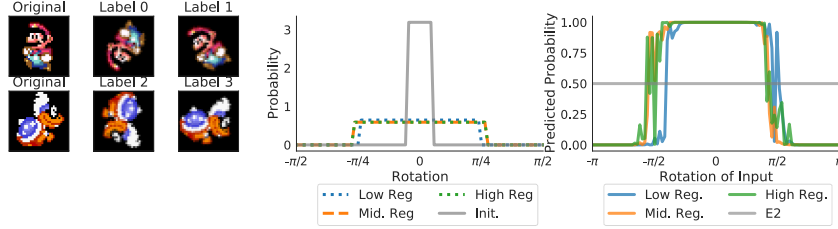

Figure 3: **Left:** Example data from the constructed Mario dataset. Labels are dependent on both the character, Mario or Iggy, and the rotation, upper half- or lower half-plane. **Center:** The initial and learned distribution over rotations. Rotations in the data are limited to $[-\pi/4, \pi/4]$ and $[-\pi, -3\pi/4] \cup [3\pi/4, \pi]$, meaning that augmenting an image by no more than $\pi/4$ radians will keep the rotation in the same half of the plane as where it started. The learned distributions approximate the invariance to rotations in $[-\pi/4, \pi/4]$ that is present in the data. **Right:** The predicted probability of label 1 for input images of Mario rotated at various angles. $E2$-steerable model is invariant, and incapable of distinguishing between inputs of different rotations.

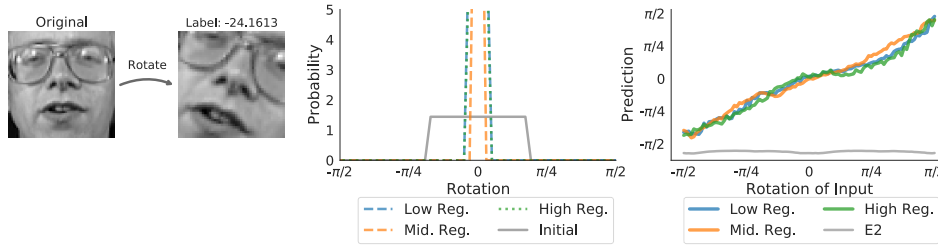

Figure 4: **Left:** The data generating process for the Olivetti faces dataset. The labels correspond to the rotation of the input image. **Center:** The initialized and learned distributions over rotations. **Right:** The predictions generated as an input is rotated. Here we see that there is no invariance present for any level of regularization - as the image rotates the predicted label changes accordingly. The $E2$-steerable network fails for this task, as the invariance to rotations prevents us from being able to predict the rotation of the image.

## 4.2 Soft Invariance: Mario & Iggy

We show that Augerino can learn *soft* invariances — e.g. invariance to a subset of transformations such as only partial rotations. To this end, we consider a dataset in which the labels are dependent on both image and pose. We use the sprites for the characters Mario and Iggy from Super Mario World, randomly rotated in the intervals of $[-\pi/4, \pi/4]$ and $[-\pi, -3\pi/4] \cup [3\pi/4, \pi]$ [33]. There are 4 labels in the dataset, one for the Mario sprite in the upper half plane, one for the Mario sprite in the lower half plane, one for the Iggy sprite in the upper half plane, and one for the Iggy sprite in the lower half plane; we show an example demonstrating each potential label in Figure 3.

In Figure 3, we see that too much rotational augmentation would make it impossible to correctly identify the pose. The limited rotations present in the data give that the labels are invariant to rotations of up to $\pi/4$ radians. Augerino learns the correct augmentation distribution within approximately 0.2 radians, and the predicted class labels follow the desired invariances, with predictions that are invariant to rotations only within subsets of $[-\pi/2, \pi/2]$.

## 4.3 Avoiding Invariance: Olivetti Faces

To test that Augerino can avoid unwanted invariances we train the model on the rotated Olivetti faces dataset [16]. This dataset consists of 10 distinct images of 40 different people. We select the images of 30 people to generate the training set, randomly rotating each image in $[-\pi/2, \pi/2]$, retaining the angle of rotation as the new label. We then crop the result to $45 \times 45$ pixel square images. We repeat the process 30 times for each image, generating 9000 training images. Figure 4 shows the data generating process and the corresponding label. Augmenting the image with any rotation would make it impossible to learn the angle by which the original image was rotated.

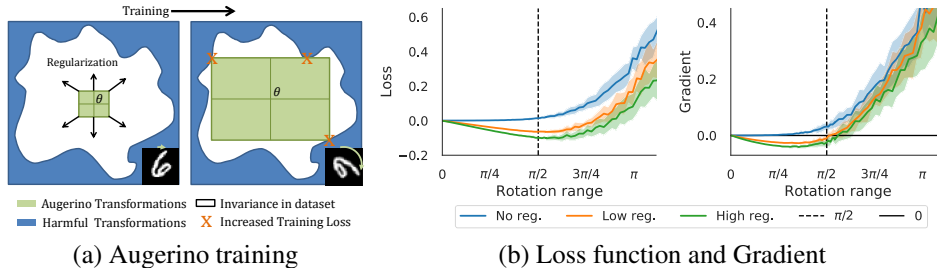

(a) Augerino training          (b) Loss function and Gradient

Figure 5: **(a):** A visualization of the space of possible transformations. Augerino expands to fill out the invariances in the dataset but is halted at the boundary where harmful transformations increase the training loss like rotating a 6 to a 9. **(b):** Loss value as a function of the rotation range applied to the input on the Mario and Iggy classification problem of Section 4.2 and its derivative. Without regularization the loss is flat for augmentations within the range $[0, \pi/2]$ corresponding to the true rotational invariance range in the data, and grows sharply beyond this range.

We find experimentally in Figure 4 that when we initialize the Augerino model such that the distribution over the rotation generating matrix $G_3$ is uniform $[0, 1]$, training for 200 epochs reduces the distribution on the rotational augmentation to have domain of support 0.003 radians wide. The model learns a nearly fixed transformation in each of the 5 other spaces of affine transformation, all with domains of support for the weights $w_i$ under 0.1 units wide.

## 5   Why Augerino Works

The conventional wisdom is that it is impossible to learn invariances directly from the training loss as invariances are constraints on the model which make it harder to fit the data [38]. Given data that has invariance to some augmentation, the training loss will not be improved by widening our distribution over this augmentation, even if it helps generalization: we would want a model to be invariant to rotations of a '6' up until it looks more like a '9', but no invariance will achieve the same training loss. However, it is sufficient to add a simple regularization term to encourage the model to discover invariances. In practice we find that the final distribution over augmentations is insensitive to the level of regularization, and that even a small amount of regularization will enable Augerino to find wide distributions over augmentations that are consistent with the precise level of invariances in the data.

We illustrate the learning of invariances with Augerino in panel (a) of Figure 5. Suppose only a limited degree of invariance is present in the data, as in Section 4.2. Then the training loss for the augmentation parameters will be flat for augmentations within the range of invariance present in the data (shown in white), and then will increase sharply beyond this range (corresponding region of Augerino parameters is shown in blue). The regularized loss in Eq. (5) will push the model to increase the level of invariance within the flat region of the training loss, but will not push it beyond the degree of invariance present in the data unless the regularization strength is extreme.

We demonstrate the effect described above for the Mario and Iggy classification problem of Section 4.2 in panel (b) of Figure 5. We use a network trained with Augerino and visualize the loss and gradient with respect to the range of rotations applied to the input with and without regularization. Without regularization, the loss is almost completely flat until the value of $\pi/2$ which is the true degree of rotational invariance in the data. With regularization we add an incentive for the model to learn larger values of the rotation range. Consequently, the loss achieves its optimum close to the optimal value of the parameter at $\pi/2$ and then quickly grows beyond that value. Figure 6 displays the results of panel (b) of Figure 5 in action; gradient signals push augmentation distributions that are too wide down and too narrow up to the correct width.

Incidentally, the Augerino solutions are substantially flatter than those obtained by standard training, as shown in Appendix F, Figure 9, which may also make them more easily discoverable by procedures such as SGD. We also see that these solutions indeed provide better generalization.

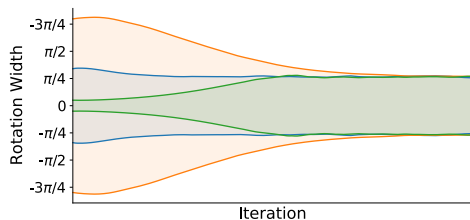

Figure 6: The distribution over rotation augmentations for the Mario and Iggy dataset over training iterations for various initializations. Regardless of whether we start with too wide, too narrow, or approximately the correct distribution over rotations, Augerino converges to the appropriate width.

# 6    Image Recognition

As Augerino learns a set of augmentations specific to a given dataset, we expect to see that Augerino is capable of boosting performance over applying any level of fixed augmentation. Using the CIFAR-10 dataset, we compare Augerino to training on data with $i$) no augmentation, $ii$) fixed, commonly applied augmentations, and $iii$) the augmentations as given by Fast AutoAugment Lim et al. [27].

Table 1: Test accuracy for models trained on CIFAR-10 with different augmentations applied to the training data.

|  | No Aug. | Fixed Aug. | Augerino (4 copies) | Augerino (1 copy) | Fast AA |
|---|---|---|---|---|---|
| Test Accuracy | 90.60 | 92.64 | $\mathbf{93.81} \pm 0.002$ | $92.22 \pm 0.002$ | 92.65 |

We compare models trained with no augmentation, a fixed commonly applied set of augmentations (including flipping, cropping, and color-standardization), Augerino, and Fast AutoAugment [27]. Augerino with $n_{copies} = 4$ provides a boost in performance with minimal increased training time. Error bars are reported as the standard deviation in accuracy for Augerino trained over 10 trials.

Table 1 shows that Augerino is competitive with advanced models that seek data-based augmentation schemes. The gains in performance are accompanied by notable simplifications in setup: we do not require a validation set and the augmentation is learned concurrently with training (there is no pre-processing to search for an augmentation policy). In Appendix F we show that Augerino find *flatter* solutions in the loss surface, which are known to generalize [30]. To further address the choice of regularization parameter, we train a number of models on CIFAR-10 with varying levels of regularization. In Figure 9 we present the test accuracy of models for different regularization parameters along with the corresponding effective dimensionalities of the networks as a measure of the *flatness* of the optimum found through training. [30] shows that effective dimensionality can capture the flatness of optima in parameter space and is strongly correlated to generalization, with lower effective dimensionality implying flatter optima and better generalization.

The results of the experiment presented in Figure 9 solidify Augerino's capability to boost performance on image recognition tasks as well as demonstrate that the inclusion of regularization is helpful, but not necessary to train accurate models. If the regularization parameter becomes too large, as can be seen in the rightmost violins of Figure 9, training can become unstable with more variance in the accuracy achieved. We observe that while it is possible to achieve good results with no regularization, the inclusion of an inductive bias that we ought to include some invariances (by adding a regularization penalty) improves performance.

# 7    Molecular Property Prediction

We test out our method on the molecular property prediction dataset QM9 [3, 34] which consists of small inorganic molecules with features given by the coordinates of the atoms in 3D space and their charges. We focus on the HOMO task of predicting the energy of the highest occupied molecular orbital, and we learn Augerino augmentations in the space of affine transformations of the atomic coordinates in $\mathbb{R}^3$. We parametrize the transformation as before with a uniform distribution for

each of the generators listed in Appendix A. We use the LieConv model introduced in Finzi et al. [14], both with no equivariance (LieConv-Trivial) and 3D translational equivariance (LieConv-T(3)). We train the models for 500 epochs on MAE (additional training details are given in C) and report the test performance in Table 2. Augerino performs much better than using no augmentations and is competitive with the hand chosen random rotation and translation augmentation (SE(3)) that incorporates domain knowledge about the problem. We detail the learned distribution over affine transformations in Appendix E. Augerino is useful both for the non equivariant LieConv-Trivial model as well as the translationally equivariant LieConv-T(3) model, suggesting that Augerino can complement architectural equivariance.

Table 2: Test MAE (in meV) on QM9 tasks trained with specified augmentation.

|  | HOMO (meV) | | | LUMO (meV) | | |
| --- | --- | --- | --- | --- | --- | --- |
|  | No Aug. | Augerino | SE(3) | No Aug. | Augerino | SE(3) |
| LieConv-Trivial | 52.7 | 38.3 | **36.5** | 43.5 | 33.7 | **29.8** |
| LieConv-T(3) | 34.2 | 33.2 | **30.2** | 30.1 | 26.9 | **25.1** |

## 8 Semantic Segmentation

In Section 3.1 we showed how Augerino can be extended to equivariant problems. In Semantic Segmentation the targets are perfectly aligned with the inputs and the network should be equivariant to any transformations present in the data. To test Augerino in equivariant learning setting we construct rotCamVid, a variation of the CamVid dataset [5, 4] where all the training and test points are rotated by a random angle (see Appendix Figure 7). For any fixed image we always use the same rotation angle, so no two copies of the same image with different rotations are present in the data. We use the FC-Densenet segmentation architecture [19]. We train Augerino with a Gaussian distribution over random rotations and translations.

In Appendix Figure 7 we visualize the training data and learned augmentations for Augerino. Augerino is able to successfully recover rotational augmentation while matching the performance of the baseline. For further details, please see Appendix B.

## 9 Color-Space Augmentations

In the previous sections we have focused on learning spatial invariances with Augerino. Augerino is general and can be applied to arbitrary differentiable input transformations. In this section, we demonstrate that Augerino can learn color-space invariances.

We consider two color-space augmentations: brightness adjustments and contrast adjustments. Each of these can be implemented as simple differentiable transformations to the RGB values of the input image (for details, see Appendix D). We use Augerino to learn a uniform distribution over the brightness and contrast adjustments on STL-10 [6] using the 13-layer CNN architecture (see Section 4). For both Augerino and the baseline model, we use standard spatial data augmentation: random translations, flips and cutout [12]. The baseline model achieves $89.0 \pm 0.35\%$ accuracy where the mean and standard deviation are computed over 3 independent runs. The Augerino model achieves a slightly higher $89.7 \pm 0.3\%$ accuracy and learns to be invariant to noticeable brightness and contrast changes in the input image (see Appendix Figure 8).

## 10 Conclusion

We have introduced *Augerino*, a framework that can be seamlessly deployed with standard model architectures to learn symmetries from training data alone, and improve generalization. Experimentally, we see that Augerino is capable of recovering ground truth invariances, including *soft* invariances, ultimately discovering an interpretable representation of the dataset. Augerino's ability to recover interpretable and accurate distributions over augmentations leads to increased performance over both task-specific specialized baselines and competing data-based augmentation schemes on a variety of tasks including molecular property prediction, image segmentation, and classification.

## Broader Impacts

Our work is largely methodological and we anticipate that Augerino will primarily see use within the machine learning community. Augerino's ability to uncover invariances present within the data, *without* modifying the training procedure and with a very plug-and-play design that is compatible with any network architecture makes it an appealing method to be deployed widely. We hope that learning invariances from data is an avenue that will see continued inquiry and that Augerino will motivate further exploration.

## Acknowledgements

This research is supported by an Amazon Research Award, Facebook Research, Amazon Machine Learning Research Award, NSF I-DISRE 193471, NIH R01 DA048764-01A1, NSF IIS-1910266, and NSF 1922658 NRT-HDR: FUTURE Foundations, Translation, and Responsibility for Data Science.

## Footnotes

[1]https://en.wikipedia.org/wiki/Augerino

[2]Mathematically speaking, this distribution is a *pushforward* by the exp map of a scaled cube with side lengths $\theta_i$ of a cube $\mu_\theta(\cdot) = \exp_* \text{Cube}_\theta(\cdot)$.

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
