[Supplementary Material]

# Appendix

Here we present additional details and experimental results. Section A gives the form of the generating matrices for the Lie group and the corresponding transformations to which they give rise. In Section B we provide details regarding the experimental setup and results in applying Augerino to image segmentation. In Section C we give the full training details for the experiments of Sections 4 and 6. In Section D we expand on the details of the color-space augmentation experiment given in Section 9 in the main text. Section E expands on the molecular property prediction experiments of Section 7, showing the learned augmentations and giving further details regarding the experimental setup. Finally Section F explains how Augerino aids in finding solutions that generalize well through looking at the *effective dimensionality* of the training solutions [30].

## A  Lie Group Generators

The six Lie group generating matrices for affine transformations in 2D are,

$$
G_1 = \begin{bmatrix} 0 & 0 & 1 \\ 0 & 0 & 0 \\ 0 & 0 & 0 \end{bmatrix}, \quad
G_2 = \begin{bmatrix} 0 & 0 & 0 \\ 0 & 0 & 1 \\ 0 & 0 & 0 \end{bmatrix}, \quad
G_3 = \begin{bmatrix} 0 & -1 & 0 \\ 1 & 0 & 0 \\ 0 & 0 & 0 \end{bmatrix},
$$
$$
G_4 = \begin{bmatrix} 1 & 0 & 0 \\ 0 & 1 & 0 \\ 0 & 0 & 0 \end{bmatrix}, \quad
G_5 = \begin{bmatrix} 1 & 0 & 0 \\ 0 & -1 & 0 \\ 0 & 0 & 0 \end{bmatrix}, \quad
G_6 = \begin{bmatrix} 0 & 1 & 0 \\ 1 & 0 & 0 \\ 0 & 0 & 0 \end{bmatrix}.
$$

(10)

Applying the exponential map to these matrices produces affine matrices that can be used to transform images. In order, these matrices correspond to translations in $x$, translations in $y$, rotations, scaling in $x$, scaling in $y$, and shearing.

## B  Semantic Segmentation: Details

In Section 8, we apply Augerino to semantic segmentation on the rotCamVid dataset (see Figure 7).

To generate the rotCamVid dataset, we rotate all images in the CamVid by a random angle, analogously to the rotMNIST dataset [23]. We note that rotCamVid only contains a single rotated copy of each image, which is not the same as applying rotational augmentation during training. When computing the training loss and test acccuracy, we ignore the padding pixels which appear due to rotating the image.

For the segmentation experiment we used the simpler augmentation distribution covering rotations and translations instead of the affine transformations (Section 3.2). We use a Gaussian parameterization of the distribution:

$$
t = (t_1, t_2, t_3) \sim \mathcal{N}(\mu, \Sigma), \quad A(t) = \begin{bmatrix} \cos(t_1) & -\sin(t_1) & 2 \cdot t_2/(w+h) \\ \sin(t_1) & \cos(t_1) & 2 \cdot t_3/(w+h) \end{bmatrix},
$$

(11)

where $\mu, \Sigma$ are trainable parameters, and $A(t)$ is the affine transformation matrix for the random sample $t$; $w$ and $h$ are the width and height of the image.

Augerino achieves pixel-wise segmentation accuracy of $69.8\%$ while the baseline model with standard augmentation achieves $68.7\%$.

## C  Training Details

**Network Training Hyperparameters**  We train the networks in Sections 4 and 6 for 200 epochs, using an initial learning rate of 0.01 with a cosine learning rate schedule and a batch size of 128.

(a) Original Data　　　(b) Augerino Sample　　　(c) Augerino Sample　　　(d) Augerino Sample

Figure 7: Augmentations learned by Augerino on the rotCamVid dataset. **(a)**: original data from rotCamVid; **(b)-(d)**: three random samples of augmentations from the learned augerino distribution. Augerino learns to be invariant to rotations but not translations.

(a) Original Data　　　(b) Augerino Sample　　　(c) Augerino Sample　　　(d) Augerino Sample

Figure 8: Color-space augmentation distribution learned by Augerino. **(a)**: original data from STL-10; **(b)-(d)**: three random samples of augmentations from the learned augerino distribution. Augerino learns to be invariant to a broad range of color and contrast adjustments while matching the performance of the baseline.

We use the cross entropy loss function for all classification tasks, and mean squared error for all regression tasks except for QM9 where we use mean absolute error.

**Train- and Test-Time Augmentations** In Algorithm 1 we include a term *ncopies* that denotes the number of sampled augmentations during training. We find that we can achieve strong performance with Augerino, with minimally increased training time, by setting *ncopies* to 1 at train-time and then applying multiple augmentations by increasing *ncopies* at *test-time*. Thus we train using a single augmentation for each input, and then apply multiple augmentations at test-time to increase accuracy, as seen in Table 1.

## D   Color-Space Augmentations: Details

In Section 9, we apply Augerino to learning color-space invariances on the STL-10 dataset. We consider two transformations:

- Brightness adjustment by a value $t$ transforms the intensity $c$ in each channel additively:

$$c' = \max(\min(c + t, 255), 0). \qquad (12)$$

  Positive $t$ increases, and negative $t$ decreases brightness.

- Contrast adjustment by a value $t$ transforms the intensity $c$ in each channel as follows[3]:

$$c' = \max\left( \min\left( \frac{259 \cdot (t + 255)}{255 \cdot (259 - t)} \cdot (c - 128) + 128, \ 255 \right), 0 \right) \qquad (13)$$

We apply brightness and contrast adjustments sequentially and independently from each other. We learn the range of a uniform distribution over the values $t$ in (12), (13). The learned data augmentation strategy is visualized in Figure 8.

Figure 9: **Top:** Test error and train loss as a function of perturbation lengths along random rays from the SGD found training solution for models. Each curve represents a different ray. **Bottom:** Test error and effective dimensionality for models trained on CIFAR-10. Results from 8 random initializations are presented violin-plot style where width represents the kernel density estimate at the corresponding $y$-value.

# E    QM9 Experiment

We reproduce the training details from Finzi et al. [14]. Affine transformations in 3d, there are 9 generators, 3 for translation, 3 for rotation, 2 for squeezing and 1 for scaling, a straightforward extension of those listed in equation 10 to 3 dimensions. Like before, we parameterize the bounds on the uniform distribution for each of these generators. We use a regularization strength of $\times 10^{-3}$.

# F    Width of Augerino Solutions

To help explain the increased generalization seen in using Augerino, we train 10 models on CIFAR-10 both with and without Augerino. In Figure 9 we present the test error of both types of models for along with the corresponding effective dimensionalities and sensitivity to parameter perturbations of the networks as a measure of the *flatness* of the optimum found through training. Maddox et al. [30] shows that effective dimensionality can capture the flatness of optima in parameter space and is strongly correlated to generalization, with lower effective dimensionality implying flatter optima and better generalization. Overall we see that Augerino enables networks to find much flatter solutions in the loss surface, corresponding to better compressions of the data and better generalization.

## Footnotes

[3] https://www.dfstudios.co.uk/articles/programming/image-programming-algorithms/image-processing-algorithms-part-5-contrast-adjustment/