[Reviews · NeurIPS 2020]

Review 1

Summary and Contributions: Post-rebuttal: I have maintained my score post-rebuttal, due to the simplicity and effectiveness of the method. I thank the authors for clarifying my issues with the submission. Pre-rebuttal: The authors propose to learn a distribution over data transformations. They do this by building differentiable data augmentation, with augmentation parameters sampled from a "uniform" distribution. The limits of the distribution are learned via the reparameterization trick on the training data. They observe improved performance on a variety of toy tasks and also on CIFAR-10 and QM9.

Strengths: The main idea is very simple and this very appealing to a broad audience of practitioners. I would certainly like to use this method in my own research. Because of the simplicity and the successful results with respect to other methods, especially in the the equivariance literature, I think this method is significant to the field. I cannot comment on the novelty of the method. The authors perform a mixture of proof-of-principle experiments demonstrating the utility of the method and that is does what they expect and then they try some harder experiments, where they also showcase beneficial performance.

Weaknesses: The mathematics, also simple, is a bit imprecise. I would tighten up the discussion of the method, specifically with claims of the linearity of the cross-entropy loss and the definition of a uniform distribution on a Lie group. I would also be careful in the discussion of equivariant functions, where it is assumed that the transformations are themselves group structured, which may not be the case (color transforms have saturation and most transformations have crops involved). The experiments could do with error bars.

Correctness: The method seems sound, as do the experimental results. If the authors still have the experimental results lying around I would like to see the standard deviation over multiple runs inserted into the results tables.

Clarity: The paper is very easy to read and written in a straight-forward manner. That said, I think when it comes to the mathematics, the authors need to be more precise. For instance what is a uniform distribution over scale? That surely depends on the parameterisation (see my addition feedback for details).

Relation to Prior Work: The related work is in general fairly covered. Perhaps an unconventional, but important work on discussion building invariance into neural networks via either imposed structure or data augmentation is Barnard, E. and Casasent, D. (1991). "Invariance and neural nets".

Reproducibility: Yes

Additional Feedback: Equation 3 - 4. It is claimed that the cross-entropy loss is linear in the output of the network. This is indeed true if the output of the network is the logits instead of the probabilities. Does it make sense to average the logits of the network over the data augmentation instead of the probabilities? I think you could remedy this by averaging over the probabilities instead of the logits and if you pull the expectation out to the front of the equation, apply Jensen's inequality, which would result in an upper bound on the loss. Line 131: why does T have to be linear for the model to be equivariant? Equation 10: shouldn't \phi \sim Q_{\theta} be \psi \sim Q_{\theta}? Line 133: do the distributions Q(\psi) and Q(\phi) have to match? Don't they just have to be be invariant to shifts on the group? That is they have to be constructed from left-invariant measures. This is sort of what you mean by uniform, I think. I guess another caveat with the equivariant model is that you have to ensure that the transformations you work with are actually group-structured. Much data augmentation is not truly group-structured because of interpolation artifacts or information destruction, for instance, digital downscaling. This should be talked about. QM9 dataset: I believe in the Finzi paper the authors train with MSE loss instead of MAE. So Im' not sure about the comparability of the results. Line 145: corresponding -> correspond


Review 2

Summary and Contributions: The submission proposes a parametrization of simple data augmentations adjustable during training. During testing, the authors propose to ensemble predictions over augmentations to make the classifier approximately invariant over the learned group.

Strengths: The submission deals with an important problem, that is, trying to infer a suitable protocol for data augmentation from the training set.

Weaknesses: I have two major concerns with this submission. First, the paper claims that prior work on data augmentation needs to know the invariance of interest a priori. However, this paper requires exactly the same thing, as the invariance of interest must be "expressable" by the learnable mapping. For instance, the authors prescribe a transformation that restrict themselves to translation, rotation, scaling and shearing invariances during training; at testing, rotation is precisely the nuisance transformation at play. Second, the proposed test-time data augmentation is a well known technique, also often used to learn equi/invariant classifiers. Examples of test-time data augmentation for better equivariance/invariance: https://arxiv.org/abs/1712.04440 https://openreview.net/pdf?id=Byxv9aioz https://arxiv.org/abs/1703.03108 https://europepmc.org/article/med/29500816 https://arxiv.org/abs/1809.01442 Test-time data augmentation is as old as AlexNet: https://papers.nips.cc/paper/4824-imagenet-classification-with-deep-convolutional-neural-networks.pdf And commonly used in ResNets: https://arxiv.org/abs/1512.03385 See "Test-time data augmentation" in the survey: https://link.springer.com/article/10.1186/s40537-019-0197-0 More formally, using Monte-Carlo over the deformations of a group to achieve invariance is a cornerstone in "Invariant Haar-Integration Kernels": https://arxiv.org/pdf/1506.02544.pdf https://arxiv.org/pdf/1612.01988.pdf Finally, it is commonly used in machine learning competitions: https://amva4newphysics.wordpress.com/2018/04/26/train-time-test-time-data-augmentation/ https://machinelearningmastery.com/how-to-use-test-time-augmentation-to-improve-model-performance-for-image-classification/ None of the references above are discussed in the manuscript. The remaining contribution of this paper is the use of the re-parametrization trick to adapt the group over which we want to be invariant on, which is in my opinion not a substantial contribution to present this paper in NeurIPS.

Correctness: Yes.

Clarity: Yes.

Relation to Prior Work: No. See weaknesses for critical omissions of prior work.

Reproducibility: Yes

Additional Feedback:


Review 3

Summary and Contributions: This paper proposes a method for learning invariances from training data. It can effectively recover distribution over augmentation procedures, while boosting performance of baseline models by adopting this learned information. The reported experimental results on several tasks demonstrate its ability on recovering invariances in a variety of scenarios.

Strengths: - The proposed method is simple and effective. It can be easily plugged and played in different NN architectures which shows its good practicality. - The experiments seem to be sufficient. The authors explored full and soft invariance learning use cases on a large variety of tasks. Although the datasets used in this paper are all with a small size, however it should be sufficient for proving the idea at the current stage. - The paper is well organized and the presentation is easy to follow.

Weaknesses: - As mentioned before, the dataset used in the experiments are all very small. It would be more convincing to see some result on medium or even large dataset such as ImageNet. But this is just a minor issue and it will not affect the overall quality of the paper. - Which model did you used in section 5 for image recognition task? To some extend it show the capability of Augerino on this task. However, on image recognition the network architecture strongly affect the result. It is interested to see what kind of chemical reaction will take place between Augrino and difference DNN architectures. --------------- after rebuttal ----------------- - Regarding the authors' response and all the other review comments I am agree with R4, that in this paper there is still some important issues needed to be re-worked before publication. I thus decided to lower my rating. I would like to encourage the authors to re-submit after revision.

Correctness: The paper is technically correct

Clarity: well written paper

Relation to Prior Work: yes

Reproducibility: Yes

Additional Feedback:


Review 4

Summary and Contributions: This paper proposes an automation process to augment the training data for a given task, such that the model would learn to apply the optimal type of augmentations in order to predict invariant or equivariant features from the data.

Strengths: - The model proposes an automation process for the augmentation of the training data, which ideally can be applied to any model leveraging data-augmentation, where the parameters of the data-augmentation can be formulated in a differentiable format. - The authors show the approach can be applied to invariant as well as equivariant data augmentation types as well as different types of data-augmentations.

Weaknesses: 1- The main assumption is that when searching for hyper-parametres, the data augmentation should not change the performance of the model obtained by f_w. How do authors know this is good? In other words if rotation by d degree is more challenging, but eventually helps the model to generalize better, this assumption would prohibit it and only allows the rotations with the exact same loss, hence reducing generalization. This is to some extent compensated through the usage of the regularization term in Eq. (6). However, the loss (without regularization) does not allow for a diverse parameter search even when it can help on generalization and the regularization term wants a diverse hyper-parameter even if this is unnecessary. How do authors find the right balance between the two? The model designer needs to find the right hyper-parameter to balance it, which is contrary to the goal of the approach, which is to automatise the hyper-parameter search. This is the major flaw of the proposed approach. 2- In Eq. (4) the reparameterization trick allows sampling within range of [a, b]. How do authors know this is a sufficient range? The reparametrization trick helps only when values ‘a’ and ‘b’ are intermediate parameters, such that gradients need to be back-propagated to some other network through ‘a’ and ‘b’. In the text and also in Fig. 1, this does not seem to be the case. How values such as ‘a’ and ‘b’ are learned to be flexible and different from the range defined in the re-parametrization trick? Also, does the initial choice of the parameters ‘a’ and ‘b’ affect the results? No experiment verifies this. 3- For segmentation experiment (Section 7), no result of the model under different rotations is shown. While experiments in Section 7 and 8 show respectively the model can be used for equivariant and channel-wise data augmentation, it is not clear what is the advantage of the proposed approach compared to a model that applies complete rotation (in Section 7) and channel-wise color transformation (in Section 8). The proposed approach would be mostly beneficial when the range of transformation should be learned, for example when not applying the transformation can reduce generalization and applying all possible transformations would take capacity of the model to learn ineffective features, hence reducing the performance when the model has a limited capacity. No experiment of such intelligent hyper-parameters learning is done. Experiments in Section 4 partially show learning the correct range of transformations, but all of them represent simple transformation when the model trainer knows how to augment (or not augment the data). In more complex tasks the right level of transformation might not be clear to the experiment designer. However, the experiments do not show any advantage of the model in such cases. This limitation is also related to point 1 mentioned above.

Correctness: The major flaws are mentioned above. In addition: T_\theta in equations (8) to (11) is not defined, neither the operation with the empty circle, so not sure if the equations (8) to (11) are correct. It is not clear how A(u) in Eq(12) is applied to the images in a differentiable way. No detail is provided. In Section 5, what kind of augmentation is applied to the model with Fixed Aug. ? If the augmentation of Fixed Aug. is more limited, then its performance would be worse.

Clarity: The writing is sometimes not clear and can be improved. Loss terms should contain the target as well, e.g. loss(prediction, target). This is the case for the equations with a loss term such as Eq. (2), (3), (4), (5). In these cases f_w(x) should be written in the equations.

Relation to Prior Work: Yes

Reproducibility: No

Additional Feedback: A lot of experimental details are missing. Even the supplementary do not contain the details of the carried out experiments. For example, it is not clear in Section 5 which type of data augmentation is applied to Fixed Aug. The authors just mention ‘commonly applied augmentations’ without specifying what the augmentations are. Such details should be provided in the text. --------------------------------------------- Rebuttal Update: I read the reviews and also the authors response. The main concern I raised above is not well-justified in the rebuttal and hence I will keep my rating. In particular, the main assumption is that the model learns to apply a set of transformations that do not hurt the performance of the model, so by default the model does not explore a set of transformations if initially they hurts the performance, but after some training time they helps the model to better generalize. For example, if rotation hurts classifying imageNet examples, even if it would help to generalize after some training time, the model does not allow such transformation. The model explores more transformations only through the regularization term in Eq. (6), which forces the model to explore more even if it hurts the performance. So these two "contradictory" components need to be balanced, through a "knowledgable" model-trainer, who knows to what extent the exploration of different values are needed. This goes contrary to the goal of the paper, which is to automatize learning the range of augmentations. So, the final range that is learned completely depends on how this hyper-parameter is set. The only cases, in which the model seems to avoid some transformations, is when the transformation is completely detrimental to the task of interest, such as the rotation experiments presented in Sections 4.2 and 4.3, where extra rotation causes wrong classification. However, if this hyper-parameter is set too high/low to motivate high exploration, it can still allow for such degenerate cases. In other tasks, where it is not detrimental, but we don't know the right range, such as in image segmentation, or classification, it is only the hyper-parameters of the regularization term that finds the range of the transformations, rather than the proposed algorithms. So, the proposed algorithm does not automatizes learning the optimal range of the hyper-parameter and still a domain-knowledgeable model-trainer should tune this value, which goes agains the goal of the paper.

[Author Response · NeurIPS 2020]

We thank the reviewers for thoughtful feedback. In this paper we introduce Augerino, a method that can automatically attain the strength of group equivariant networks through learning data augmentations. Augerino is a flexible method that can be combined with any pre-existing architecture. We note that Augerino does not require one to specify invariances a priori, merely a group transformation. We show Augerino is capable of **learning** augmentations, from training data alone, which boosts accuracy when applied at test time over a wide range of tasks. We want to emphasize that Augerino is a distinctive contribution: other works on data augmentation are not geared towards learning equivariances at training time and would not even directly apply to most of the applications considered in our paper.

**R1:** Thank you for the constructive review, we appreciate the feedback and will incorporate the suggestions accordingly. With respect to the discussion of the mathematics, we commit to making the mathematical statements in Sections 3.1 and 3.2 more precise, we will switch from the notation of the transformation and distribution of $T_\phi$,$Q_\theta$ to $g, \mu$. Our use of uniform distribution should be made more precise, and that it could depend on the parameterization if not careful. We will make it clear that the distribution is meant to be the Haar measure on the group, and that the appropriate condition is left translation invariance $d\mu(hg) = d\mu(g)$. The transformation $T_\phi$ needs to act linearly so that we can pull it out of the expectation in Eqs. $9-10$. Unlike with learning invariances, the extension to equivariance in Section 3.1 does require the group structure and is not applicable to more general set of transformations which may lack inverses or closure, and we will add this point of discussion in the text. While this requirement is an idealization and limitation in some ways (cropping, boundary effects), we feel that this simplification is justified. Note that the invariant model only requires an invariant measure, but not necessarily the existence of inverses, and thus includes transformation semi-groups. We will mention the deep scale-spaces paper in this context. Additionally, thank you for the pointer to the Barnard and Casasent paper, this is a great early example of the utility of invariance.

**R2:** While we appreciate the pointers to test-time augmentation, which we are happy to reference, we want to be clear that these papers are complementary to Augerino and do not diminish our contribution. Indeed, most of these works would not even directly apply to the applications we consider. Augerino is distinctive in that it **learns** an appropriate distribution over augmentations at **train-time**. We motivate the utility of Augerino in Section 4, demonstrating both why one may want to learn a distribution over augmentations, rather than just applying them.

We also note that for Augerino to be applicable we need only the invariance to be represented by a group transformation. In Section 4 we do know the correct invariances, but these experiments serve to show that when we do know the correct distribution over invariances we can recover the ground truth, and are in no way meant to show the limiting case performance of the method. In Sections 5 and 6 we do not have a known target set of invariances yet Augerino outperforms competing methods. While in some cases the set of invariances will never be known, if we do believe that invariance is a trait we wish our model to have then group transformations are a promising path forward as they encompass a broad set of transformations that has been shown to improve performance on many well studied tasks; for examples see [6, 7, 8, 12].

**R3:** We thank the reviewer for the constructive comments and ideas for extensions. The model used in Section 5 is an 8 layer deep convolutional neural network with a max channel width of 256. We will be sure to include the details of all architectures in the appendix of the camera ready.

We have not observed any evidence that Augerino is susceptible to loss of performance in using alternate network architectures. One of the key strengths of Augerino is its ability to be combined freely with any model. Throughout the experiments we use a range of different convolutional and feed forward neural networks, chosen only to be appropriate for the given task not for their compatibility with Augerino. In each case (the demonstrations of Section 4, and the benchmark improvements of Sections $5-8$) Augerino performs as expected and improves accuracy over baselines.

Finally, we will incorporate larger datasets in the camera-ready submission. Preliminary results suggest Augerino will perform well on larger datasets such as CIFAR100, however we avoid presenting any incomplete results here.

**R4:** Thank you for your thoughtful comments. *Assumption of hyperparameters:* The parameters $a$ and $b$ are *learned parameters of the model* and not hyperparameters; the reparametrization trick gives us gradients with respect to these parameters. Therefore $a$ and $b$ can learn to accommodate an arbitrary range, and we do not find performance sensitive to their initialization. We will clarify. *Learning transformations:* as discussed in response to reviewer 2, the experiments in Section 4 highlight our ability to uncover the ground-truth augmentations. In later sections we show Augerino can outperform competing methods in which desired transformations are unknown.

*Remaining points:* The augmentation in Eq. 12 is done through computing the exponential map as the solution to a differential equation which allows back-propagation to flow through the affine transformations. In Section 5 the fixed augmentation indicates a standard set of transformations applied to the training data , we do expect fixed augmentation to perform worse, that is the central motivation to this work. We will be sure to include these details and clarify all notation in Section 3 in the camera-ready.

[Meta-Review · NeurIPS 2020]

Authors propose to learn a method for data-augmentation that improves performance as compared to data-augmentation strategy where all parameters are randomized. The results are not on datasets used by SOTA methods and some of them are in the appendix instead of the main paper. I agree with the authors that R2 might have misunderstood the paper and did not participate in the post-rebuttal discussion. It is also mentioned in his review, "The remaining contribution of this paper is the use of the re-parametrization trick to adapt the group over which we want to be invariant on, which is in my opinion not a substantial contribution to present this paper in NeurIPS." I don't think we should judge papers solely based on novelty in the technical section. While the idea is indeed simple, it does lead to performance improvements and does bring to forth the importance of learning how to augment data, instead of just performing random data-augmentation. R4's main concern seem to be requirement of hyper-parameter tuning, but I believe most methods require it. This cannot solely be the reason for rejection. R1 who recommends acceptance, says, "The main idea is very simple and this very appealing to a broad audience of practitioners. I would certainly like to use this method in my own research." While the idea does not produce ground-breaking results, I can see it being impactful in a wide-variety of problems. I broadly agree with R1. Despite negative scores from the reviewers and discussion with other ACs, I recommend the paper be accepted.